# SARS-CoV-2 Genotyping Highlights the Challenges in Spike Protein Drift Independent of Other Essential Proteins

**DOI:** 10.3390/microorganisms12091863

**Published:** 2024-09-09

**Authors:** Jeremy W. Prokop, Sheryl Alberta, Martin Witteveen-Lane, Samantha Pell, Hosam A. Farag, Disha Bhargava, Robert M. Vaughan, Austin Frisch, Jacob Bauss, Humza Bhatti, Sanjana Arora, Charitha Subrahmanya, David Pearson, Austin Goodyke, Mason Westgate, Taylor W. Cook, Jackson T. Mitchell, Jacob Zieba, Matthew D. Sims, Adam Underwood, Habiba Hassouna, Surender Rajasekaran, Maximiliano A. Tamae Kakazu, Dave Chesla, Rosemary Olivero, Adam J. Caulfield

**Affiliations:** 1Office of Research, Corewell Health, Grand Rapids, MI 49503, USA; martin.witteveen-lane@corewellhealth.org (M.W.-L.); hosam.farag@corewellhealth.org (H.A.F.); sanjana.arora@corewellhealth.org (S.A.); charitha.subrahmanya@priorityhealth.com (C.S.); david.pearson@corewellhealth.org (D.P.); austin.goodyke@corewellhealth.org (A.G.); mason.westgate@corewellhealth.org (M.W.); surender.rajasekaran@corewellhealth.org (S.R.); david.chesla@corewellhealth.org (D.C.); 2College of Human Medicine, Michigan State University, Grand Rapids, MI 49503, USA; bharga15@msu.edu (D.B.); vaugha53@msu.edu (R.M.V.); frischau@msu.edu (A.F.); baussjac@msu.edu (J.B.); bhattih1@msu.edu (H.B.); tcook1697@gmail.com (T.W.C.); jacksonmitchell3431@gmail.com (J.T.M.); maximiliano.tamaekakazu@corewellhealth.org (M.A.T.K.); rosemary.olivero@helendevoschildrens.org (R.O.); 3Advanced Technology Lab, Corewell Health, Grand Rapids, MI 49503, USA; sheryl.alberta@corewellhealth.org (S.A.); samantha.pell@corewellhealth.org (S.P.); 4Genetics and Genome Sciences Program, BioMolecular Science, Michigan State University, East Lansing, MI 48824, USA; ziebajac@msu.edu; 5Section of Infectious Diseases, Corewell Health, Royal Oak, MI 48073, USA; matthew.sims@corewellhealth.org; 6Department of Internal Medicine, Oakland University William Beaumont School of Medicine, Auburn Hills, MI 48309, USA; 7Division of Mathematics and Science, Walsh University, North Canton, OH 44720, USA; aunderwood@walsh.edu; 8Adult Infectious Disease, Corewell Health, Grand Rapids, MI 49503, USA; habiba.hassouna@corewellhealth.org; 9Division of Pulmonary and Critical Care Medicine, Corewell Health, Grand Rapids, MI 49503, USA; 10Pediatric Infectious Disease, Helen DeVos Children’s Hospital, Corewell Health, Grand Rapids, MI 49503, USA; 11Microbiology Lab, Corewell Health, Grand Rapids, MI 49503, USA

**Keywords:** SARS-CoV-2, COVID-19, viral genome, drift, Spike, nonstructural protein, papain-like protease, cysteine protease

## Abstract

As of 2024, SARS-CoV-2 continues to propagate and drift as an endemic virus, impacting healthcare for years. The largest sequencing initiative for any species was initiated to combat the virus, tracking changes over time at a full virus base-pair resolution. The SARS-CoV-2 sequencing represents a unique opportunity to understand selective pressures and viral evolution but requires cross-disciplinary approaches from epidemiology to functional protein biology. Within this work, we integrate a two-year genotyping window with structural biology to explore the selective pressures of SARS-CoV-2 on protein insights. Although genotype and the Spike (Surface Glycoprotein) protein continue to drift, most SARS-CoV-2 proteins have had few amino acid alterations. Within Spike, the high drift rate of amino acids involved in antibody evasion also corresponds to changes within the ACE2 binding pocket that have undergone multiple changes that maintain functional binding. The genotyping suggests selective pressure for receptor specificity that could also confer changes in viral risk. Mapping of amino acid changes to the structures of the SARS-CoV-2 co-transcriptional complex (nsp7-nsp14), nsp3 (papain-like protease), and nsp5 (cysteine protease) proteins suggest they remain critical factors for drug development that will be sustainable, unlike those strategies targeting Spike.

## 1. Introduction

SARS-CoV-2 currently ranks as the most sequenced virus in human history, with nearly 9 million genomes sequenced as of early 2024 within NCBI [1] and 16.8 million in GISAID [2], the two largest repositories of viral genomics. Within NCBI Virus, most (~97%) of the sequenced SARS-CoV-2 genomes were isolated from oronasopharynx swabs, with less than one percent from feces and blood (Figure 1A).

The early genomic datasets suggested that the virus could accumulate one substitution every 11 days, with extensive focus on how changes in the genome could alter the viral spread to inform the pandemic response [3]. Frequent reports of genomic changes have contributed to the public perception that the virus is rapidly changing, a perception that has influenced scientists to focus on genomic change rather than conservation. Although extensive work addressed the changes to the Spike (S) protein and how they result in immune escape [4], the virus contains additional proteins coded by the genes ORF1ab (ORF1a and ORF1b), ORF3a, E, M, ORF6, ORF7a, ORF8, N, and ORF10 [5] that have remained poorly characterized for variant impacts. SAR-CoV-2 genomes contain each of these proteins at similar detection as Spike (Figure 1B), but assessments of protein-coding changes within them have been less of a focus throughout the literature [6].Based on the NCBI, the majority of our genomes for SARS-CoV-2 were generated in Europe and North America (Figure 1C), with the USA, United Kingdom, Germany, Denmark, and Switzerland depositing the most genomes as countries (Figure 1D). Within the USA, many of the states with the largest populations generated the most genomes (Figure 1E), including California (CA), Florida (FL), and Texas (TX). Several states sequenced a higher number of genomes relative to their population (Figure 1F), including Vermont (VT), Massachusetts (MA), and Minnesota (MN). As there is a clear disbalance in genotyping of SARS-CoV-2 relative to geography, there has also been a lag in the publications of data analysis for these genotypes from diverse geographies, emphasizing individual groups to define dynamics of the virus in unique locations [7,8,9,10]. Yet, there has been a lack of strategy for exploring the full viral genome drift to proteins outside Spike.

The SARS-CoV-2 genomics field has been centered around phylogenetics and “high-impact” variants [11], which was beneficial for predicting new risk strains in the pandemic response. As data accumulated over the years, the extensive volumes of genomic data, the complexities of sequencing bias, the challenge of rare variants to “potential” risks, and the geographical complexities [12] have challenged the genomic utility of the massive sequencing investments during the pandemic. As the field transitions into the endemic phase of SARS-CoV-2, we must strategize how our investments in genomics can highlight aspects of targeted treatment dynamics and understand selective pressures between viral survival and virulence within each geographical region.

This paper combines our deep evolutionary and structural dynamics of SARS-CoV-2 [13] with two years of genotyping within a community hospital system. The analysis integrates the current state of SARS-CoV-2 genomic and structural knowledge, elucidating opportunities to move forward in the COVID-19 endemic era that other teams can utilize to transition from phylogenetics into protein impacts and drug targeting.

## 2. Materials and Methods

This work was approved by the Corewell Health West Institutional Review Board (IRB #CHW 2020-470). Nasal swabs were obtained from routine clinical testing at Corewell Health regional hospitals and collection sites throughout Michigan (MI), United States. Samples positive for SARS-CoV-2 (cycle threshold Ct value ≤ 32) were extracted using a QIAcube HT (Qiagen, Hilden, Germany) with a QIAamp 96 Viral RNA kit with Offboard Lysis 1NS-2 protocol. Illumina’s COVIDSeq Test kit was used for sequencing in rounds of 96-well plates consisting of 94 samples, a positive control, and a no template control for each run. IDT for Illumina DNA/RNA UD Index sets A through D (96 indexes per set) were used to identify each unique sample. Individual libraries were pooled per plate, quantified, and then normalized to 4 nm. Four normalized libraries were pooled together and sequenced on the NextSeq 500 using a mid-output sequencing cartridge. FASTQ analysis was performed using Illumina’s DRAGEN COVID Lineage application, and the flattened FASTA file for each sample was exported to NCBI and GISAID with collection dates for further protein annotations. NCBI processing of sample uploads as the routine part of NCBI virus annotations yielding updated Pangolin and protein annotations [14].

FASTA files for each SARS-CoV-2 protein of all sequenced samples were exported from NCBI Virus tool and aligned with the Kalign algorithm [15] using UGENE tools [16] with the gap open penalty set to 54.95, gap extension penalty at 8.52, terminal gap penalty at 4.42, a bonus score of 0.02, and the standard genetic code option. Only missense variants were assessed for the sequenced samples, with all synonymous changes of the viral genomes not analyzed in the current study. Following alignments within the UGENE tool, the consensus of all aligned sequences for each protein was calculated using the default consensus mode set to a threshold of 10%. The consensus sequence was added to each alignment as the top listed sequence, and alignments were transformed into matrix positioning using Mega software (version 5) [17]. Within the matrix, any sequences with uncertain base calls (N to ? annotation in FASTA files) over the protein resulted in that entire protein sequence being removed, yielding a highly curated alignment with uncertain error removed. After sequence removal, a unique list of amino acids at each position was calculated from the matrix, and the percent of sequences with the same amino acid as the consensus was calculated, with one minus that number serving as the drift rate for each amino acid.

The Protein Data Bank (PDB) [18] was quired on 3/14/2024 for all structures deposited containing the species “SARS-CoV-2”. Results were extracted for the per-molecule species annotations and manually curated for the protein annotations of human and SARS-CoV-2 molecules to fix uncommon annotation use. The Spike-ACE2 integrated structure [13], nsp7-14 complex (PDB file 7egq) [19], nsp3-ISG15 (6xa9) [20], and nsp5-Nirmatrelvir (8b2t) [21] files were used to map variants using YASARA software (version 24) [22]. In short, each molecule of SARS-CoV-2 was homology modeled from the source PDB file using the protein consensus sequence above. Coloring of amino acid drift was done using YASARA python coding. Data for all USA and international genotypes were extracted for SARS-CoV-2 (taxon ID 2697049) using NCBI Virus on 24 July 2024. Each variant identified in structures was annotated to lineages using the NCBI Virus mutation tool.

## 3. Results

Corewell Health clinical sites are in a mixed community of a city (Grand Rapids) with a regional triage center and additional hospitals that service areas of rural Michigan, USA. The state of Michigan has ranked as one of the top geographical areas for SARS-CoV-2 genomes (Figure 1E, red), with a median density of genomes relative to the population size (Figure 1F, red). Corewell Health is an extensive hospital system that serves over 1.3 million health plan members. Corewell Health sequenced 7518 SARS-CoV-2 genomes from December 2021 through December 2023 that passed quality control with >90% coverage and a median of 500 reads (Figure 2A). These samples were all used to process genome annotations with the NCBI standard SARS-CoV-2 workflow. There were a few delta strains throughout the two years of sample/data collection and sequencing, but most of the sequencing was of various omicron annotations (Figure 2B). The genotyping data were 82% concordant with national data (USA) over the study period based on a per genotype per month annotation of NCBI SRA data relative to Corewell Health samples (Figure 2C). This suggests that our genotyping data represent similar trends to national data for month-over-month sampling and genotype calls, opening the door for variant and protein analyses using our subset of data.

As large volumes of samples (in the millions) are computationally limiting due to the volume, random sampling is often used to assess viral drift that is not centered on ultra-rare variant detection. Utilizing four months throughout the sampling, we further compared the breakdown of Pangolin annotations at Corewell Health relative to all samples in the state of Michigan, the USA, and any deposited sample (internationally) to determine the utility of a single hospital system analysis. The absence of alpha and low levels of delta observed early in the study are in concordance with the level of USA cases (Figure 2D). Corewell Health accurately reflects the genotyping in the state of Michigan and USA, while the international data do have some difference in timing, likely the result of migration of genotypes being different over geographical regions month-to-month. In the case of this work, the high correlation of the Corewell Health month-to-month genotype levels is in high agreement with NCBI USA data and thus represents a sampling strategy for variants arising over lineages and not focusing on the ultra-rare changes that occurred throughout the pandemic.

Each protein of SARS-CoV-2 was aligned at an amino acid level and assessed for coverage and drift relative to the consensus sequence (Figure 3). Extensive parsing of ambiguous protein sequences was used to remove uncertain sequences, resulting in variability for each protein coverage. The ORF1ab protein was broken into the various protein cleavage products nsp1-nsp16. A total of 166,741 protein sequences for 9634 amino acids were assessed, yielding 64,979,205 positions assessed in our genotyping. The envelope protein (E) has a drift rate of 1.21% per amino acid, followed by ORF6a (0.90%) and Spike (S, 0.76%), representing the proteins with the most changes relative to the consensus sequence of each protein. Many of the nonstructural proteins have a low drift rate, including nsp7 (0.01%), nsp10 (0.02%), and nsp5 (0.02%). Spike (S) has 20 different amino acids with >10% drift from the consensus, which, when normalized for protein size, is the highest along with ORF6a (Figure 3A). Proteins such as nsp2, nsp5, nsp7, nsp8, nsp9, nsp10, nsp14, nsp16, ORF7a, ORF8, and ORF10 had no amino acid variants with >10% frequency (Figure 3B). A total of 43 amino acids has a drift rate of >10% relative to the consensus (Table 1). Spike, ORF6a, E, nsp1, and nsp6 had the highest frequency of variants over 10% of samples.

To identify variant outcomes relative to known protein structures, we curated the known structural knowledge of SARS-CoV-2 within the RCSB PDB as of 14 March 2024 (Figure 4). The RCSB PDB is the main repository for all solved protein structures [23]. A total of 4501 SARS-CoV-2 unique molecules have had a structure determined, including 2878 human proteins bound to them (Figure 4A). Other species bound are primarily based on antibody molecules produced in non-human organisms. Of the SARS-CoV-2 molecules with structures, Spike is the most prevalent, followed by the two proteases nsp3 and nsp5 (Figure 4B). Of the human-bound proteins, the majority were antibodies or MHC complexes (Figure 4C), with a few Spike-ACE2 structures and several large ribosome complexes.

Structure files were used to align the genomic variants of SARS-CoV-2 for Spike-ACE2 (Figure 5A), the multi-protein nsp7-nsp14 SARS-CoV-2 co-transcriptional complex (Figure 5B), nsp3-ISG15 (Figure 5C), and nsp5- nirmatrelvir (PAXLOVID, Figure 5D) structure files. Spike has multiple variants within the ACE2 interaction region (Figure 5A). A cluster of variants at amino acids 495, 498, and 502 contact ACE2 but maintain functional conservation. Amino acid 495 uses arginine (R) or glutamine (Q), while 498 uses tyrosine (Y) or asparagine (N), all of which maintain hydrogen bonding to ACE2 sites. Amino acid 502 uses histidine and tyrosine, maintaining aromatic ACE2 pi orbital interactions. Additional functional conservation can be found for ACE2 interactions at amino acids 400 (R or K), 490 (Q or R), 453 (F, L, or V), and 483. Amino acid 483 transitions throughout the two years of sequencing from phenylalanine (F) to valine (V) to proline (P), which all maintain the hydrophobic ACE2 interactions. This variant also correlates to changing lineage annotations with this as position 486 of BA.5.1 with a valine and XBB.1.5 with a proline. The proline has previously been suggested as a variant of concern due to risks in increasing binding affinity to ACE2 [24,25].

In contrast to the high drift of Spike, the entire multi-protein SARS-CoV-2 co-transcriptional complex (Figure 5B) only contains two variants of potential alteration impact. The nsp12 amino acid 671 transitions from a glycine (G) to a serine (S) in early 2023, allowing for the flexible bend angle near the RNA contact points. This variant (RDRP: G671S) represents 2% of the GenBank deposited samples from 2/2024 to 7/2024 with the highest frequency in un-assignable and GE.1.2.1 lineages according to the NCBI Virus mutation tool. The most impactful change within the complex occurs at nsp13 amino acid 392, where in early 2022, it changed from an arginine (R) to cysteine (C). This location falls near the nsp7 and nsp8 interaction sites and thus could potentially alter interactions. Within GenBank, 99% of deposited samples from 2/2024 to 7/2024 contain this change over all major annotated lineages.

The top studied drug target sites are the proteases nsp3 (Figure 5C) and nsp5 (Figure 5D) of ORF1ab, where no variants above 1% of our sequenced samples are found at the targetable active sites. Those variants that are above 1% all fall on nonfunctional sites such as 822, 1001, and 1039 of nsp3 or 24, 90, or 132 of nsp5.

## 4. Discussion

As any virus continues to drift and change over time, there is a critical need to focus on the biological outcomes of the changes rather than over-focusing on genotypes. Although genotypes can inform the kinetics of viral propagation, yielding insights into how a virus’s geographical and temporal movements happen [26], they can also result in confusion and fear when at the forefront of the conversation about a pandemic or endemic. As the COVID-19 pandemic became so focused on genotypes and the rapid changes of SARS-CoV-2, the general population and policymakers often over-reacted [27,28]. As the current work shows, many SARS-CoV-2 proteins have had few changes outside Spike, with even fewer that are functional relative to our known structural knowledge and dynamics [13] of the virus. At the onset of the pandemic, groups such as ours called for drug development based on shared function with other Coronaviruses [13,29,30]. However, international investments in Spike-based therapies far outpaced that of other protein complexes. Although viruses such as HIV have high levels of protein changes throughout based on replication dynamics that make drug development complex [29], more focus should be placed on genotypes’ structural impacts, molecular outcomes, and drug-targeting dynamics for each virus.

A simple word search of relevant terms to this paper shows that surface glycoprotein and Spike terms far outnumber any of the other proteins of SARS-CoV-2 (Table 2). The general terms for the virus, such as COVID-19 and SARS-CoV-2, are higher in Google relative to both PubMed and Google Scholar. Surface glycoprotein is the highest returned term within PubMed and far outpaces other proteins in both PubMed and Google Scholar, suggesting that, as noted above, there is an overfocus on surface glycoprotein (Spike, S) that does not match the genotyping conservation discussed within the current work. A search of “SARS-CoV-2 mutations” and “Spike” returns 4920 returns on Google Scholar, while “SARS-CoV-2 mutations” and “ORF” or “nsp” returns only 929 and 598 hits. Both ORF and nsp represent multiple proteins, further highlighting the contrast relative to Spike. This highlights a fundamental issue with our current genotyping interpretation and overfocus on genomic changes relative to conservation analysis.

It is well known that Spike continues to drift, which has led to the iterative development of changes to COVID-19 vaccines, a likely never-ending process based on the current genotyping insights. The changes in Spike were a major factor for the elevation of pathogenicity for SARS-CoV-2 over other coronaviruses [31], and several of the amino acid changes observed in Omicron Spike also modulate ACE2 and immune evasion [32,33]. As our data show, what is less appreciated is the functionally conserved changes happening at the ACE2 binding site, while no other protein shows such levels of functional drift. The data suggest that pressure for Spike changes is not found in other SARS-CoV-2 coded proteins. These insights are highly relevant to moving forward in the virus’s treatment and public health dynamics in the endemic phase, suggesting a significant need to transition from Spike-based therapies to other proteins. Others have noted that this drift makes it very challenging also to target the Spike-ACE2 interaction with drugs [34]. As we continue to engage the arms race of Spike antibody targeting with Spike drift for ACE2 interactions, it is feasible to drive selective pressure to elevate Spike dynamics that could enhance virulence, viral dynamics, and increase binding affinity [35]. This suggests pivoting to non-Spike therapies and strategies [36].

So why does Spike have amino acid drift at a rate so different from many other viral proteins? The answer is likely that the surface proteins need not be maintained for a virus to replicate; they are only required for spreading [37,38,39]. All a virus needs the surface proteins to do is evade immune detection and get inside a cell, and there are many proteins they can find to get inside a cell with many ways to evade immune detection. Viruses are always finding new cell epitopes with which to integrate. As the protein sequences change on surface proteins like Spike, there may be some selection of binding surfaces, as we have shown for ACE2-Spike interactions in SARS-CoV-2, while changes also enhance immune evasion and confirm selective benefit [40]. It is uncertain and unlikely vaccinations impact drift of the surface proteins as this drift is already naturally occurring. Even without vaccinations, the freedom of the surface protein to change due to its limited role in viral replication, the proteins behave much as any protein with reduced selective pressure [41]. If the protein changes enough, selection finds another protein partner in the host to enter a cell, as evolution has shown often in large evolutionary changes that give rise to new viral species. This is a similar concept throughout evolution when large genomic segments duplicate and reduce the selective pressure of one of the two copies of paralogs to give radiant evolution potential [42]. However, when the sequence changes for the inner machinery of proteins in a virus, like the proteases and polymerases, the virus cannot survive and the change is quickly lost to subsequent viral replications due to selective pressures [43]. In the process of the constant viral turnover, these pressures on protein drift can be seen and are likely the result of protein-centric genotyping insights from our work.

A genome is only a coding material for the biological functions manifested from protein structures. In virology, the focus has often been the DNA/RNA sequence, as the material is more straightforward to observe and make sense of. That gives rise to assessments of biology using phylogenetics and explaining viral drift at DNA/RNA level where the genotypes make it appear a virus rapidly changes. As the work here shows, the drift of DNA often looks chaotic, while the proteins look stable. Thus, the use of genomes in drug design has a view of chaos and constant change. As most of our drugs target proteins, the assessment of viral drift requires a protein-centric assessment. The research field did this well throughout the pandemic for Spike, but the rest of the SARS-CoV-2 proteome was left overlooked.

One of the most prolific examples of protein structural drug design is that of HIV, where the use of structures and the rapid drift have opened up a toolbox of drugs [44]. As viruses such as HIV rapidly change due to the reverse transcriptase process, these viruses highlight how challenging it is to treat, where focus on the conserved mechanisms that, when drifted, impact the viral function are strategic targets of our emerging drugs [45]. As viruses often share these critical protein mechanisms, the drugs that target conserved protein pockets can often open rapid treatment approaches for emerging viruses, as was so well done in targeting the proteases of SARS-CoV-2 that originated from the structure-based design within SARS-CoV [46]. The structure-based drug design can be seen emerging in dengue [47], Zika [48], Hepatitis [49], and influenzas [50], to name a few.

Our data are supported by the real-world continued and growing success of the antiviral medication nirmatrelvir-ritonavir (brand name Paxlovid) and nsp5-targeting, whose efficacy has not been impacted by the two years of viral drift [51,52]. Our early work in viral evolution for SARS-CoV-2 relative to other coronaviruses suggested the same outcomes [13], and the current genotyping data also highly support non-Spike therapies. It also suggests that some T-cell receptor and antibody-active proteins, such as the nucleocapsid, may represent more sustainable models for public health vaccinations.

As epidemiology begins to decipher the long-term risks of SARS-CoV-2, including in non-severe COVID-19 [53], and as the virus becomes endemic, similar to Epstein Barr virus with rare risks to populations [54], we need to consider the role of viral drift heavily in our future therapy strategies against SARS-CoV-2. We must also enhance our knowledge of proteins that are not drifting and conserved across multiple Coronaviruses. For example, the role of NSP3 and other proteins in immune suppression [29], which remain unchanged throughout the SARS-CoV-2 drift and are present in many patients’ blood responses [55], yet have largely been ignored and underfunded. Suppose we were to invest as heavily into these unknown mechanisms that remain highly conserved within the SARS-CoV-2 proteins as we did for Spike and vaccine strategies. In that case, it is likely that we would transform not only the knowledge and treatments for COVID-19 but also make major strides for numerous other viruses that share mechanisms. Continuous integrated surveillance systems and research are necessary to monitor the virus’s evolution, evaluate the effectiveness of vaccines and treatments, and rapidly respond to emerging variants to ensure public health safety.

## Figures and Tables

**Figure 1 microorganisms-12-01863-f001:**
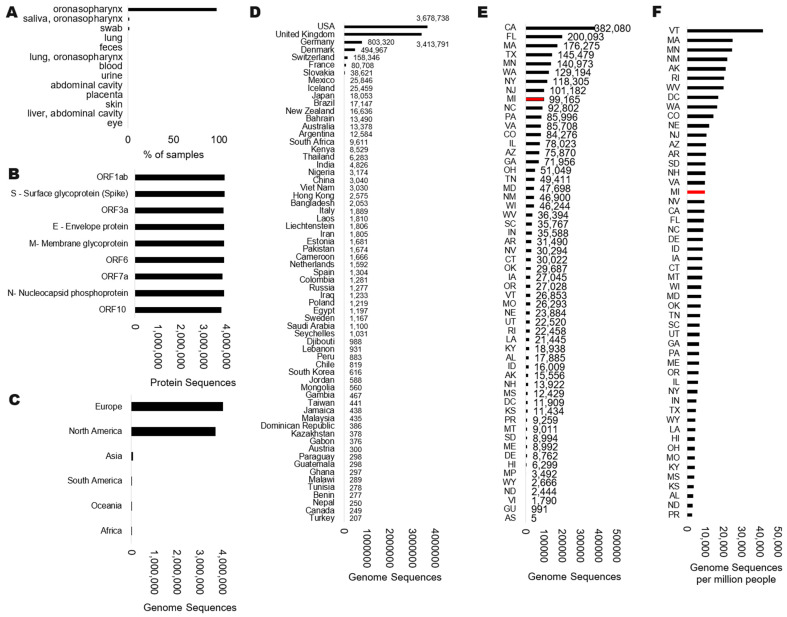
Status of SARS-CoV-2 genomes (taxid:2697049) within NCBI Virus. Data extractions were performed on 24 July 2024. Data are shown for the isolate (**A**), protein annotations (**B**), or geographical region based on continent (**C**), country (**D**), or state within the USA (**E**). Panel (**F**) shows the normalized genome sequences in each state of the USA relative to the million individuals within the state based on the 2020 census levels. Red boxes indicate the location of current genotyping data.

**Figure 2 microorganisms-12-01863-f002:**
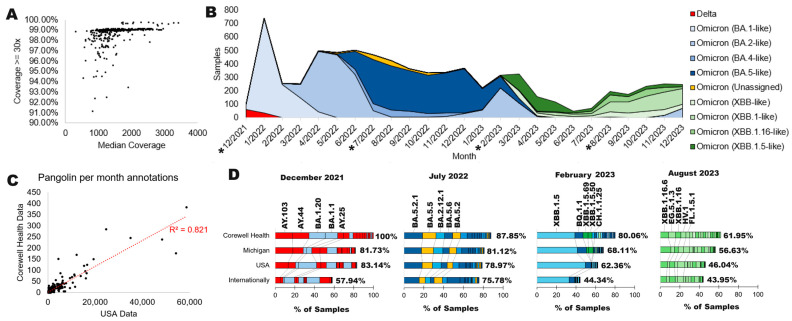
Corewell Health COVID-19 sequencing data. (**A**) Sequencing statistics for samples analyzed at Corewell Health using Illumina chemistry. (**B**) SARS-CoV-2 genotypes per month from December 2021 through December 2023. The “*” indicates months with further processing in panel D. (**C**) Correlation analysis of the Pangolin annotation each month from the USA (with <100 ambiguous N’s per sequence) relative to Corewell Health. (**D**) Bar plots for the percent of samples each of four months (starred in panel (**D**)) for data from Corewell Health, the state of Michigan, the USA, and any NCBI deposited sample (Internationally). Data are shown with the percent of the top 20 Pangolin annotations (top 5 labeled), with the % listed next to the stacked plot representing how many of the total samples are covered by the top 20. Colors correspond to panel (**B**) annotations for higher-level genotypes.

**Figure 3 microorganisms-12-01863-f003:**
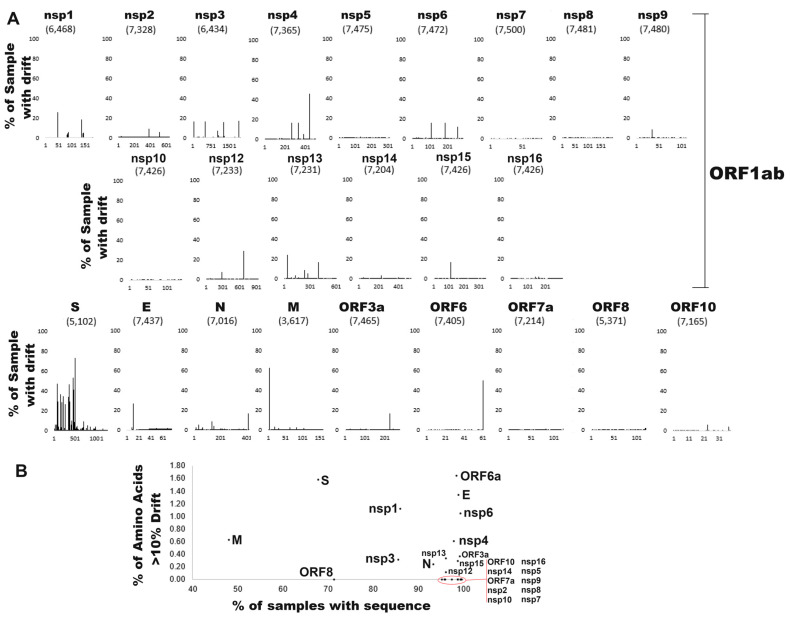
SARS-CoV-2 protein drift. (**A**) Per protein amino acid drift relative to the consensus sequence. Shown next to each protein name is the number of sequences without ambiguity. The x-axis shows the amino acid number of each protein, and the y-axis shows the % of samples with drift relative to the consensus amino acid. (**B**) The per protein sequence coverage relative to the number of amino acids with >10% variation.

**Figure 4 microorganisms-12-01863-f004:**
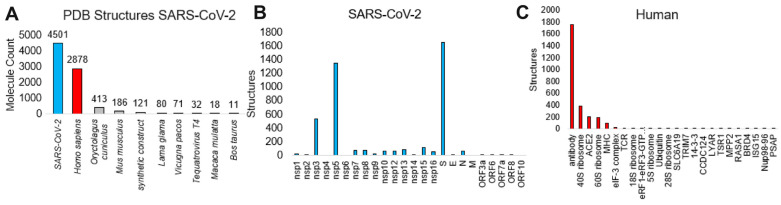
Protein structure analysis of SARS-CoV-2. (**A**) The number of SARS-CoV-2 protein structure molecules for various species in the Protein Databank (PDB) as of 14 March 2024. (**B**) The number of unique structures for each of the SARS-CoV-2 proteins (cyan). (**C**) The number of human structures (red) interacting with SARS-CoV-2 proteins.

**Figure 5 microorganisms-12-01863-f005:**
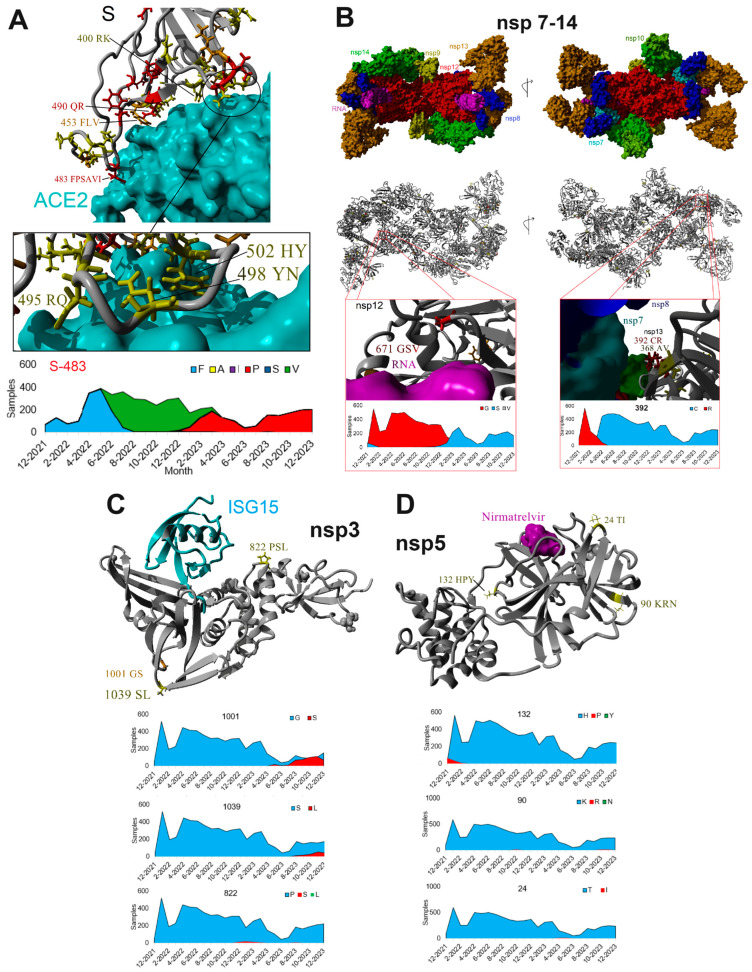
Genomic drift mapped to SARS-CoV-2 protein structures. (**A**) Analysis of variants in Spike (S) at the interface of ACE2 interaction. (**B**) Analysis of variants for the SARS-CoV-2 co-transcriptional complex. (**C**) Analysis of variants for the nsp3 papain-like protease with the human ISG15 interaction site. (**D**) Analysis of variants for the nsp5 SARS-CoV-2 protein with nirmatrelvir (PAXLOVID) bound.

**Table 1 microorganisms-12-01863-t001:** Amino acids that have the highest drift rate from each protein’s consensus sequence.

Protein	Complete Sequences for Protein	Amino Acid Number	Consensus	Amino Acids Used	Amino Acids Used	% Drift
surface glycoprotein	5102	483	F	6	FPSAVI	73.11
membrane glycoprotein	3617	3	D	4	DHNG	62.37
surface glycoprotein	5102	449	R	5	RLMWQ	53.21
ORF6	7405	61	L	5	LIDFP	49.99
surface glycoprotein	5102	66	H	5	H-YFS	47.53
surface glycoprotein	5102	67	V	3	V-I	47.43
surface glycoprotein	5102	343	R	5	RTISK	46.37
NSP4	7365	438	L	3	LFY	46.06
surface glycoprotein	5102	457	N	5	NKISY	41.40
surface glycoprotein	5102	141	Y	2	Y-	36.50
surface glycoprotein	5102	443	G	5	GSTDV	35.59
surface glycoprotein	5102	210	G	4	GERV	34.77
surface glycoprotein	5102	336	D	5	DHVYG	33.83
surface glycoprotein	5102	442	V	8	VPSHALIF	31.145
surface glycoprotein	5102	143	H	7	HQK-YPL	29.60
surface glycoprotein	5102	487	F	3	FSP	29.38
surface glycoprotein	5102	80	V	2	VA	29.09
surface glycoprotein	5102	365	L	2	LI	29.09
surface glycoprotein	5102	180	Q	6	QEVHKL	28.68
NSP12	7233	671	G	3	GSV	28.61
surface glycoprotein	5102	249	G	3	GVD	26.81
envelope protein	7437	11	T	3	TAM	26.45
NSP1	6468	47	K	2	KR	25.57
surface glycoprotein	5102	490	Q	2	QR	24.52
NSP13	7231	36	S	2	SP	24.31
NSP1	6468	135	R	4	RSKN	18.69
NSP3	6434	1892	A	3	ATG	17.28
NSP4	7365	264	F	2	FL	17.05
NSP4	7365	327	I	4	ITVF	17.03
ORF3a	7465	223	I	2	IT	17.03
nucleocapsid phosphoprotein	7016	410	R	5	RLSHC	17.00
NSP15	7426	112	I	3	ITN	16.89
NSP13	7231	392	C	2	CR	16.60
NSP3	6434	24	I	2	IT	16.52
NSP3	6434	489	S	2	SG	16.47
NSP3	6434	38	K	2	KR	15.99
NSP3	6434	1265	S	2	S-	15.96
NSP3	6434	1266	L	3	LIV	15.96
NSP6	7472	105	L	2	LF	15.91
NSP6	7472	186	I	2	IV	15.79
NSP6	7472	257	L	3	LFH	12.17
surface glycoprotein	5102	441	K	5	KTNRM	11.00
surface glycoprotein	5102	453	F	3	FLV	10.80

**Table 2 microorganisms-12-01863-t002:** A search for relevant words from SARS-CoV-2 in various databases. Each database was searched on 24 July 2024.

	PubMed (All)	PubMed (Since 2020)	Google Scholar (All)	Google Scholar (Since 2020)	Google (All)	Ratio Google to PubMed (Since 2020)
surface glycoprotein	788,636	92,484	1,840,000	26,700	45,000,000	487
Membrane glycoprotein	786,289	91,863	1,920,000	24,200	38,400,000	418
COVID-19	438,172	292,493	5,040,000	1,020,000	6,400,000,000	21,881
pandemic	274,892	171,320	4,220,000	683,000	1,510,000,000	8814
SARS-CoV-2	231,386	186,717	2,500,000	781,000	459,000,000	2458
viral genome	117,243	19,241	4,220,000	30,400	120,000,000	6237
viral genotype	64,224	10,674	2,130,000	16,700	23,500,000	2202
Envelope protein	52,391	5939	2,110,000	23,900	46,300,000	7796
RNA-directed RNA polymerase	38,009	4142	35,100	7190	40,200,000	9705
Helicase	36,622	7834	332,000	25,200	19,500,000	2489
Nucleocapsid	28,311	5146	208,000	27,500	7,970,000	1549
surface glycoprotein spike	9146	2269	91,900	18,200	1,210,000	533
papain like protease	2031	574	95,100	17,000	776,000	1352
3C like protease	1580	986	850,000	16,300	8,810,000	8935
Nucleocapsid phosphoprotein	1452	694	19,700	6350	131,000	189
nsp2	1180	302	20,800	8450	459,000	1520
nsp1	1124	324	26,600	12,400	428,000	1321
nsp3	1049	377	20,500	11,700	269,000	714
nsp4	949	203	17,000	6880	208,000	1025
ORF1ab	676	448	17,900	17,400	307,000	685
ORF6	542	131	16,700	7070	192,000	1466
nsp12	483	329	10,600	8830	470,000	1429
nsp5	478	174	11,600	6780	212,000	1218
ORF3a	380	240	8840	8040	145,000	604
nsp14	301	169	7940	6430	235,000	1391
nsp13	256	154	7280	6080	540,000	3506
nsp10	254	136	6640	4640	176,000	1294
2′-O-methyltransferase	237	65	130,000	18,500	16,800,000	258,462
nsp16	230	123	6880	5690	169,000	1374
nsp15	215	118	6030	4910	224,000	1898
nsp7	205	114	6920	5260	189,000	1658
nsp9	200	96	5350	3500	189,000	1969
nsp8	191	116	6850	5170	136,000	1172
ORF10	190	67	8350	4450	73,900	1103
ORF7a	189	114	5350	4830	96,300	845
Guanine-N7 methyltransferase	175	34	2230	1070	53,700	1579
nsp6	170	89	7660	4880	152,000	1708
Uridylate-specific endoribonuclease	86	58	745	480	17,200	297

## Data Availability

All sequencing data annotations are available at https://doi.org/10.6084/m9.figshare.25988050.v1. The drift calculation for each amino acid is located at https://doi.org/10.6084/m9.figshare.26364514.v1.

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
