# Peer review of "SARS-CoV-2 Genotyping Highlights the Challenges in Spike Protein Drift Independent of Other Essential Proteins"

_microorganisms, 2024, doi:10.3390/microorganisms12091863_

Round 1
Reviewer 1 Report
Comments and Suggestions for Authors
Prokop et al studied the drift of protein structures of SARS-CoV2 in genomic scale and attempted to emphasize development of non-spike protein mediated drug development. They sequenced 7518 SARS-CoV2 genomes from Michigan and the sequence was used to model the protein structures from PDB database. It is true that functional consequences of proteins would be better informative for drug development.
1. The enormous effort is needed to centralize the abundant genomic sequence of SARS-CoV2 in NCBI and GSAID in different geographical regions of the world to find the drift of a particular protein from its consensus sequence. Authors used the sequence from Corwell health of Michigan which a very low representation of overall SARs-CoV2 genome sequences. They should show/write an explanation why this conclusion could represent all SARS-CoV2 (omicron).
2. Authors mentioned that Corwell health sequenced 7518 SARs-CoV2 genomes from December, 2021 to December, 2023. Authors should mention how many are used for this study. I am wondering why there was so less delta strain and no alpha strain when delta had major peaks in 2022 and alpha was roaming in 2021 December? Did authors select few deltas and omicron only?
3. The material and methods need to be more elaborated to understand every step of protein structure modelling from SARS-CoV2 genomic sequence. For example, some principles of calculation of drift rate for each amino acid from consensus sequence to a pool of genomic sequences. It would be also useful to write in brief the mechanism of each software they used to do the structure modelling from vast genomic sequences.
4. Authors should also mention whether and how the software filters synonymous amino acid changes?
5. Authors should explain what sequences of each SARS-CoV2 protein they used to derive consensus sequences?
6. Writings within the figures are illegible, especially in Figure1, 3 and 5. Authors should manually rewrite them with larger script and better resolution.
7. In discussion, the author should discuss elaborately why determining structural drift could be used as a better alternative for drug development.

Author Response
We thank the reviewer for the outstanding suggestions, all of which have been incorporated into the newest version of the manuscript. Below are detailed responses to each point.
Comment 1.1- The enormous effort is needed to centralize the abundant genomic sequence of SARS-CoV2 in NCBI and GSAID in different geographical regions of the world to find the drift of a particular protein from its consensus sequence. Authors used the sequence from Corwell health of Michigan which a very low representation of overall SARs-CoV2 genome sequences. They should show/write an explanation why this conclusion could represent all SARS-CoV2 (omicron).R1. In figure 2C, the Pangolin annotation shows a good correlation of the SARS-CoV-2 sequences at Corewell Health with the lineage reported in the USA.
Response 1.1- We thank the reviewer for pointing this out. We have expanded our text on describing Figure 2C and why this is a critical factor in allowing our data to scale. We have added a Figure 2D to further break down genotype comparisons to USA and international data and added a paragraph to describe the new insights.
Comment 1.2- Authors mentioned that Corwell health sequenced 7518 SARs-CoV2 genomes from December, 2021 to December, 2023. Authors should mention how many are used for this study. I am wondering why there was so less delta strain and no alpha strain when delta had major peaks in 2022 and alpha was roaming in 2021 December? Did authors select few deltas and omicron only?
Response 1.2- We have added a sentence in the results to specify further that all samples were processed, We have also added the Dec 2021 comparison in Figure 2D to directly address the reviewer, highlighting that our sampling of delta and lack of alpha was in agreement with data in the USA.
Comment 1.3- The material and methods need to be more elaborated to understand every step of protein structure modelling from SARS-CoV2 genomic sequence. For example, some principles of calculation of drift rate for each amino acid from consensus sequence to a pool of genomic sequences. It would be also useful to write in brief the mechanism of each software they used to do the structure modelling from vast genomic sequences.
Response 1.3- We have added more details throughout the methods section as suggested.
Comment 1.4- Authors should also mention whether and how the software filters synonymous amino acid changes?
Response 1.4- We have added this into the methods, “Only missense variants were assessed for the sequenced samples, with all synonymous changes of the viral genomes not analyzed in the current study.”
Comment 1.5- Authors should explain what sequences of each SARS-CoV2 protein they used to derive consensus sequences?
Response 1.5- This has been detailed now in the methods section.
Comment 1.6- Writings within the figures are illegible, especially in Figure1, 3 and 5. Authors should manually rewrite them with larger script and better resolution.
Response 1.6- We have increased the size of the figures to make text more legible, which the image size is ultimately up to the journal for final figure size. For Figure 3 and 5, we also changed the spacing to enable a larger panel.
Comment 1.7- In discussion, the author should discuss elaborately why determining structural drift could be used as a better alternative for drug development.
Response 1.7- We have added two paragraphs into the discussion for why structural biology is critical to understanding drift of proteins and why this is so important for drug design.
Reviewer 2 Report
Comments and Suggestions for Authors
This is an interesting paper that has plenty of data, which is both a strength and weakness. With such large amount of data, the reader is able to pore through the data to look for trends. Conversely, it becomes somewhat more difficult for the reader and authors to come up with concrete conclusions. There are a few questions I have.
1) Why is the S so heavily mutated compared to other proteins? Is it because it is the main target of vaccines? Or is it a natural process since S is one of the few exposed proteins in CoV and therefore a natural target of the immune system? Have you seen any differences in the data from different countries especially between those that are heavily vaccinated and less vaccinated?
https://www.ncbi.nlm.nih.gov/pmc/articles/PMC9834082/
2) During the Wuhan outbreak, Shang et al did a computational study showing that SARS2 S-ACE2 affinity is 1000 times greater than that of SARS1. People immediately concluded that that is why SARS2 is so much more infectious than SAR2. In your data, were you able to see the weakening of the S-ACE2 binding affinities in Omicron. If so, how do you reconcile the fact that Omicron is still very infectious given that it is endemic.
https://www.nature.com/articles/s41586-020-2179-y
3) Pastorio et al observed certain trends in the mutations of S."Individual mutations of S371F/L, S375F, and T376A in the ACE2-receptor-binding domain as well as Q954H and N969K in the hinge region 1 impaired infectivity, while changes to G339D, D614G, N764K, and L981F moderately enhanced it. Most mutations in the N-terminal region and receptor-binding domain reduced the sensitivity of the Spike protein to neutralization by sera from individuals vaccinated with the BNT162b2 vaccine and by therapeutic antibodies". Were you able to reproduce their observations and add to the trends that they were seeing?
https://www.ncbi.nlm.nih.gov/pmc/articles/PMC9289044/
4) Fastq should be changed to the correct nomenclature, FASTQ. Similarly Fasta should be corrected to FASTA.
5) Datum (singular) is/was. DATA (plural) are/were
Author Response
We thank the reviewer for their thoughtful review. Below are point-by-point responses.
Comment 2.1- Why is the S so heavily mutated compared to other proteins? Is it because it is the main target of vaccines? Or is it a natural process since S is one of the few exposed proteins in CoV and therefore a natural target of the immune system? Have you seen any differences in the data from different countries especially between those that are heavily vaccinated and less vaccinated?
Response 2.1- We have added a paragraph into the discussion to address why Spike can change so much quicker that other proteins and how this fits into the broader viral evolutionary biology.
Comment 2.2- During the Wuhan outbreak, Shang et al did a computational study showing that SARS2 S-ACE2 affinity is 1000 times greater than that of SARS1. People immediately concluded that that is why SARS2 is so much more infectious than SAR2. In your data, were you able to see the weakening of the S-ACE2 binding affinities in Omicron. If so, how do you reconcile the fact that Omicron is still very infectious given that it is endemic.
Response 2.2- We have added the following sentence, “The changes in Spike were a major factor for the elevation of pathogenicity for SARS-CoV-2 over other coronaviruses [31], and several of the amino acid changes observed in Omicron Spike also modulate ACE2 and immune evasion [32,33].”
Comment 2.3- Pastorio et al observed certain trends in the mutations of S."Individual mutations of S371F/L, S375F, and T376A in the ACE2-receptor-binding domain as well as Q954H and N969K in the hinge region 1 impaired infectivity, while changes to G339D, D614G, N764K, and L981F moderately enhanced it. Most mutations in the N-terminal region and receptor-binding domain reduced the sensitivity of the Spike protein to neutralization by sera from individuals vaccinated with the BNT162b2 vaccine and by therapeutic antibodies". Were you able to reproduce their observations and add to the trends that they were seeing?
Response 2.3- As addressed in response 2.1 and 2.2, we have added additional text about variants in Spike and how important they are. Nothing in our current work would be able to reproduce the previous observations of functionality of the changes that others have done.
Comment 2.4- Fastq should be changed to the correct nomenclature, FASTQ. Similarly Fasta should be corrected to FASTA.
Response 2.4- This has been corrected in the manuscript.
Comment 2.5- Datum (singular) is/was. DATA (plural) are/were
Response 2.5- We wish to point out that the international standards define data as singular or plural as “data” can be interpreted at ontological levels and the term datum is becoming less frequent in the field because of this issue. The integration of information into data represents a singular form, just like the word database or big data are treated in a singular form. The use of data throughout represents the aggregate of information, making it singular and thus the proper use “data is”. All of the text was analyzed using grammar services (Grammarly) that agreed with this usage.
Round 2
Reviewer 1 Report
Comments and Suggestions for Authors
The paper is now much improved.
Reviewer 2 Report
Comments and Suggestions for Authors
Improvement seen